# ADHD and Mental Health Symptoms in the Identification of Young Adults with Increased Risk of Alcohol Dependency in the General Population—The HUNT4 Population Study

**DOI:** 10.3390/ijerph182111601

**Published:** 2021-11-04

**Authors:** Anders Dahlen Forsmo Lauvsnes, Mette Langaas, Alexander Olsen, Jasmin Vassileva, Olav Spigset, Rolf W. Gråwe

**Affiliations:** 1Department of Mental Health, Faculty of Medicine and Health Sciences, Norwegian University of Science and Technology, 7491 Trondheim, Norway; rolf.w.grawe@ntnu.no; 2Outpatient Department, NKS Kvamsgrindkollektivet AS, 7036 Trondheim, Norway; 3Department of Mathematical Sciences, Faculty of Information Technology and Electrical Engineering, Norwegian University of Science and Technology, 7491 Trondheim, Norway; mette.langaas@ntnu.no; 4Norwegian Computing Center, SAMBA, 0373 Oslo, Norway; 5Department of Psychology, Faculty of Social and Educational Sciences, Norwegian University of Science and Technology, 7491 Trondheim, Norway; alexander.olsen@ntnu.no; 6Department of Physical Medicine and Rehabilitation, St. Olavs University Hospital, 7006 Trondheim, Norway; 7Institute for Drug and Alcohol Studies, Virginia Commonwealth University, Richmond, VA 23219, USA; jasmin.vassileva@vcuhealth.org; 8Department of Psychiatry, Virginia Commonwealth University, Richmond, VA 23219, USA; 9Department of Clinical and Molecular Medicine, Faculty of Medicine and Health Sciences, Norwegian University of Science and Technology, 7491 Trondheim, Norway; olav.spigset@ntnu.no; 10Department of Clinical Pharmacology, St. Olav University Hospital, 7006 Trondheim, Norway; 11Division of Psychiatry, Department of Research and Development, St. Olavs University Hospital, 7006 Trondheim, Norway

**Keywords:** public health, alcohol, ADHD, cognitive dysfunction, mental health, classification

## Abstract

Symptoms of ADHD are strongly associated with alcohol use disorders, and mental health symptoms attenuate this relationship. There is limited knowledge about how specific symptoms of inattentiveness and hyperactivity/impulsivity can explain this association. We aimed to identify self-reported executive cognitive functioning and mental health and variables that may help identify subjects with an elevated risk of alcohol dependence in the general population. Data included 3917 subjects between 19 and 30 years old in the 4th Trøndelag Health Study. The Adult ADHD Self report Scale—Screener, the Hospital Anxiety and Depression Scale, and demographic variables were used as input variables. The alcohol screening instrument CAGE was used as the response variable for binary alcohol dependence risk. We used logistic regression and automated model selection to arrive at our final model that identified sex, age, inattentiveness, hyperactivity/impulsivity symptoms, and anxiety as predictors of having a CAGE score ≥2, achieving an area under the receiver operating characteristic curve of 0.692. A balanced accuracy approach indicated an optimal cut-off of 0.153 with sensitivity 0.55 and specificity 0.74. Despite attrition in the data, our findings may be important in the assessment of individual risk for alcohol dependency and when developing algorithms for risk triage in public health.

## 1. Introduction

Substance use disorders are associated with drug-related incentive salience, reward deficits, and compromised cognitive functioning [1]. Combined with enhanced incentive salience, cognitive dysfunction may have a detrimental effect on the inhibition of alcohol and substance use behaviour [2]. Executive dysfunction is a central cognitive dysfunction that may influence the risk of alcohol problems and produce a loss of top-down cognitive control and increased impulsivity [3]. Specifically, attenuated inhibitory control is closely related to impulsivity [4].

Meta-analyses and population studies have shown that attention deficit-hyperactivity disorder (ADHD) is strongly associated with lifetime alcohol use [5,6]. Even with subthreshold diagnostic symptomatology, the presence and severity of ADHD symptoms are strongly associated with substance abuse [7]. These associations also persist when controlling for age, sex, socioeconomic class, ethnicity, and a lifetime history of conduct, major depressive or any anxiety disorder [8]. There are indications that ADHD typology seems to be shared between alcohol and other types of substance use problems [9]. Impulsivity in individuals with substance abuse behaviours may stem from pre-existing traits, substance use effects such as maladaptive plasticity and neurotoxicity, and likely incentive or reward sensitivity [10].

Executive dysfunctions in the form of inhibitory problems and impulsivity are core symptoms of ADHD. Impulsivity has been conceptualized as a transdiagnostic marker of psychiatric symptoms in general and substance use specifically and it is likely that impulsivity moderates the effects of internalising symptoms on substance use [3]. In addition, within-person variability in impulsivity may increase the odds of both heavy drinking and alcohol-related problems [11]. Despite some bias towards clinical populations in the current research literature, several studies have also demonstrated significant relationships between impaired response inhibition or impulsivity and high-risk alcohol use in non-clinical populations [12]. 

Bozkurt and colleagues [13] found that depression severity and self-reported trait impulsivity predicted alcohol use severity in treatment-seeking patients, but that depression was no longer a significant predictor when controlling for ASRS symptom severity. The relationship between anxiety disorder and alcohol use may be due to substance-induced anxiety (e.g., withdrawal), self-medication of stress, or a common vulnerability [14]. 

Males and females have approximately the same rate of initiating substance use, but somewhat different trajectories concerning the pace of escalation, substances used, and comorbidities. This may be related to differences in the development trajectories of the central nervous system [15].

There is a need for more knowledge about factors associated with increased odds of alcohol dependence in the general population. In this study, we aimed to identify neurocognitive variables affecting the risk for alcohol dependence in a large sample general population while controlling for mental health symptoms, by using well known brief self-report measures of externalising symptoms of executive dysfunctions related to hyperactivity/impulsivity and inattentiveness as well as internalising symptoms such as anxiety and depression. In particular we aim to help further the understanding of the ADHD-related neurocognitive mechanisms leading to variations in the probability of alcohol abuse and in line with the findings of Fuller-Thomson and colleagues [16], and as such, facilitate the development of more person-centered and early intervention in public health in line with the findings of Fuller-Thomson and colleagues [16].

## 2. Materials and Methods

### 2.1. Participants and Procedures

The HUNT study (the Trøndelag Health Study) constitutes an extensive population database for health-related research containing questionnaires, physical examination findings, and biological specimens. Four public health surveys of the general adult population in the Nord-Trøndelag County, Norway, have been completed between 1984 and 2019. We used data from the 4th wave of the HUNT Study (HUNT4). HUNT4 consisted of several questionnaires, one general (Q1) and one age cohort specific (Q2), in addition to sub-studies with separate questionnaires, specimens, and measurements. HUNT 4 was carried out between 2017 and 2019.

For the current study, data from Q1 and Q2 were used. These were merged based on unique personal identification numbers. Q1 could be completed by hand or by the web. The participants that completed Q1 by hand went to a research field station to hand them in and were invited to fill out a second questionnaire (Q2). The participants who completed the web based Q1 were invited to the field station to complete Q2 there. Those who did not show up at the field stations did not provide data in Q2. 

All 17,500 inhabitants of Nord-Trøndelag County between 19 and 30 years old were invited to participate. The current data set included an age cohort ranging from 19 to 30 years old at the time of participation that had completed and returned Q1 (*n* = 6510). Subsequently, 6123 participants were invited for Q2, of whom 3917 provided data. Attrition beyond Q1 (*n* = 2593) occurred as subjects either not completing Q2 despite being available and asked to do so (*n* = 2206) or did not show up at the field-testing station (*n* = 387). The data does not contain information about race or ethnicity.

The average annual alcohol consumption in Norway is lower than in the EU [17]. Norwegian men typically drink more and have a riskier alcohol consumption than women. Moreover, young adults typically have a more high-risk drinking pattern, characterized by binge drinking with an increased frequency of consuming ≥6 units per drinking occasion. For the age interval 25–34 years, 83% drank alcohol in the last 12 months with a mean of 33 occasions, and 60% had at least one occasion of ≥6 drinks, with the mean number of ≥6 occasions being 8 per year [18].

### 2.2. Measures

#### 2.2.1. Executive Dysfunction—Adult ADHD Self-Report Scale Screener—ASRS

The six-question World Health Organization Adult ADHD Self-Report Scale Screener (ASRS-6) Screener is a brief self-report instrument that has demonstrated the ability to discriminate DSM-IV cases of ADHD from non-cases [19]. Variables corresponding to part A Screener of the ASRS version 1.1 (ASRS-6) included in Q2 of HUNT4 were used as a proxy measure of executive dysfunction. The ASRS symptom score is also significantly correlated with general impulsivity measures such as the short-form Barrat Impulsiveness Scale, BIS-11 [13]. The ASRS-6 has been validated both in normal populations and in treatment-seeking alcohol use populations [13]. The six-items version has also been validated in the Norwegian language for substance use populations as part of the full instrument [20]. The ASRS-6 typically loads onto the two factors, inattentiveness (ASRS IA, items 1–4) and hyperactivity/impulsivity (ASRS H/I, items 5 and 6) [21]. We generated composite scores for factors 1 (ASRS IA, Cronbach’s α = 0.75) and 2 (ASRS H/I, Cronbach’s α = 0.49) by averaging the corresponding items; the possible range of score was 0 to 4. A total of 176 subjects missed one or more items on the ASRS and were excluded (155 cases for ASRS 1 and 153 for ASRS 2). We then created two new predictor variables from the averages of ASRS 1 and ASRS 2. To avoid collinearity, the total sum score of the two composite scores was not used as input in regression analyses.

#### 2.2.2. Mental health—Hospital Anxiety and Depression Scale—HADS

The Hospital Anxiety and Depression Scale (HADS) is a frequently used questionnaire for assessing anxiety and depression symptoms [22]. The instrument has seven questions about anxiety (HADS-A) and seven questions about depression (HADS-D), with each item having a range from 0 to 3 on an ordinal scale, with a possible total score of 42. This numeric score was used as a predictor variable in the regression analysis. The scale has been validated for use in alcohol use-dependent populations [23] and in the general medical population [24]. We did not use clinical cut-off scores.

#### 2.2.3. Alcohol Dependence—CAGE

The CAGE questionnaire is a screening instrument of high-risk alcohol behaviour and was used as an outcome variable in this study. There are four questions, asking the subjects if they (i) have ever felt the need to cut down on their drinking, (ii) have been annoyed by other criticizing their drinking, (iii) have felt guilty due to their drinking, and (iv) have taken a drink the next morning to cope with anxiety or withdrawal. Every question can be answered by “yes” (1 point) or “no” (0 points) [25]. In a population study, CAGE’s internal reliability was adequate across age and sex cohorts [26]. The CAGE has been thoroughly validated in adults and is considered suitable for identifying patients at risk for alcohol dependence. However, there is some indication that it may be less sensitive and specific when applied to young adults in the general population [27].

We created a binary response variable (CAGE2). Subjects with a score of 2 or more were considered cases; this is a commonly accepted cut-off for being at high risk of alcohol dependence [28]. We replaced a total of 25 cases with missing items in CAGE, by setting CAGE equal to 0 for all subjects having one missing and at least 3 ‘No’ or all missing values in CAGE and responded that they did not drink alcohol in the background questions in Q1, as these participants would not be able to reach the lower cut-off score of 2. A total of 475 subjects were identified as cases with high-risk alcohol behaviour using CAGE ≥2. Using 1 as cut-off would have yielded 975 as possible cases, 3 would have yielded 325, and 4 would have yielded 49. A CAGE score of 2 was kept as cut-off, as the instrument has already been found to possibly be over-sensitive in this age cohort [27].

### 2.3. Statistical Analyses

#### 2.3.1. Statistical Software

We performed all analyses using R statistical tool version 3.6.3 [29].

#### 2.3.2. Analyses of Attrition

To analyse attrition from Q1 to Q2, between-group comparisons were used, both t-tests to look at differences between participants and non-participants, chi-square test for proportions of sex between Q1 and Q2 participants, and logistic regression with participation in Q2 as binary response and sex and the total number of alcohol units consumed, obtained from Q1 as explanatory variables.

#### 2.3.3. Model Selection

A logistic regression model was fitted to all data, with the HADS Anxiety and Depression sub scores and the ASRS sub scores, as well as Sex and Age as candidate input variables and CAGE2 as the response variable. We used a best subset model selection with the Akaike Information Criterion (AIC) to identify the best model from the candidate set of explanatory variables. Only main effects were considered. The R statistical package “glmulti” was used for automated model selection [30]. A first-order Taylor approximation was employed to provide confidence intervals for the odds ratios (ORs).

#### 2.3.4. Evaluating the Classifier

To evaluate model performance, we used 10-fold cross-validation to calculate the Receiver Operating Characteristics (ROC) curve and the area under the ROC curve (ROC-AUC). The pROC R-package was used for these calculations [31]. For imbalanced binary data (the size of the case group is substantially smaller than the control group) classification methods may be biased towards the most frequent class. Balanced accuracy was used as a measure to give a balanced view of both the sensitivity and the specificity of the classification [32]. Balanced accuracy is the arithmetic mean of sensitivity and specificity (sensitivity + specificity)/2) at a given cut-off. Our data is unbalanced, with only about 12.1% (475) having a CAGE score of 2 or more (and as such being classified as ‘case’).

## 3. Results

Descriptive statistics of the predictor variables are found in Table 1. Female subjects had higher average anxiety (*p* = 7.14 × 10^−23^) and depression scores (*p* = 0.03), whereas male subjects had higher average hyperactivity/impulsivity scores (*p* = 5 × 10^−9^).

The sex proportions in respondents changed significantly from Q1 to Q2, from 56.5% women in Q1 to 63.5% women in Q2 (χ^2^ = 55.417, df = 1, *p* = 0.04). Descriptive statistics of input variables in the modelling and analysis of sex differences are shown in Table 1. There were modest, albeit significant, differences between men and women concerning anxiety and hyperactivity/impulsivity symptoms. Correlation analyses between input variables revealed moderate to strong correlations between measures of internalising and externalising symptoms.

After AIC model selections, the final logistic regression model included five predictors, and the fitted model results are shown in Table 2. Male sex was the most important predictor (OR 3.14, 95% CI 2.54–3.90), whereas age was negatively predictive of CAGE2, albeit with a low odds ratio (OR 0.96, 95% CI 0.93–1.00). Executive dysfunction, as measured by both ASRS IA (OR 1.25, 95% CI 1.08–1.44) and ASRS H/I (OR 1.22, 95% CI 1.09–1.38), was positively predictive of the CAGE2 score. Anxiety, measured by HADS, was also left as a significant predictor in the final model, with a modest odds ratio (OR 1.08, 95% CI 1.05–1.11) Although CAGE2 was weakly correlated to HADS Depression (0.102, df = 3775, *p* = 8.55 × 10^−11^), HADS Depression was not included in the final model, because a model not including HADS Depression obtained a more favourable AIC. This final model used 3672 complete rows of data.

The model shows an overall acceptable ability to identify subjects with a high risk of alcohol dependence with an AUC of 0.692. The ROC curve is presented in Figure 1.

For an illustration of the model’s ability to discriminate between CAGE2 and non-CAGE2 subjects, we calculated sensitivity and specificity at different cut-off thresholds (Table 3). A threshold of 0.153 gave the highest balanced accuracy, with a value of 0.65 and a corresponding sensitivity of 0.55 and specificity of 0.74.

To further illustrate the working of the model and help in the interpretation of the parameters we used the estimated model to predict the probability of CAGE score ≥ 2 for five different hypothetical persons with varying values in the predictor variables in Table 4. We can see that being a 20-year-old male with an average ASRS IA and H/I score of 3 and 4 respectively and an anxiety score of 20 on HADS has a predicted probability of about 0.666 of having a CAGE score ≥ 2.

## 4. Discussion

In this cross-sectional population study of young adults, we show that being male, having lower age, more executive dysfunction in the form of ADHD-symptoms, and increased anxiety levels increase the probability of being at risk for alcohol dependence. In line with other prior studies (e.g., [8]), male sex was the strongest predictor. Both self-reported inattentive and hyperactivity/impulsivity symptoms were significant positively associated with having higher odds of alcohol dependence. Our findings are important to consider when assessing the individual risk for alcohol use problems in the clinic and developing algorithms for such assessment both for clinical and population level public health work. It is also of clinical importance to better understand the role of self-reported executive dysfunctions as a transdiagnostic marker in the expression of substance use disorders in general. Depression was not included in our final study model, indicating that the effect of depression on alcohol use is moderated by executive dysfunction. This finding is in line with previous research [33].

Our model classified subjects with a high risk of alcohol dependence (defined by a CAGE score of ≥2), achieving ROC-AUC of 0.692. The model produced higher sensitivity than specificity at the 0.05 and 0.10 cut-off thresholds. This may be desirable when the goal is to identify and target interventions to at-risk subjects. However, in low-risk populations, the cost will be a high proportion of false positives, i.e., subjects being classified as being at risk but not developing alcohol dependency. For the optimal cut-off level to achieve as high balanced accuracy (average of sensitivity and specificity) as possible, we identified a cut-off about 0.153, giving a sensitivity of 0.55 and specificity of 0.74. We included five hypothetical subjects for illustration purposes that both may be helpful in interpreting our findings and may illustrate its possible usefulness both at an individual and group level, by showing what different values in the predictor variables mean for the predicted probabilities of CAGE score of ≥2.

ROC-AUC values of up to 0.87 have been reported in the classification of subjects with either heroin and amphetamine dependence in protracted abstinence [34]. Their study used a substantial number of input variables, both self-reported measures of internalising and externalising symptoms and behavioural tests of cognitive functioning in a verified drug-dependent population. Our approach is considerably less resource-demanding and may be more rational for primary identification of young adults at particular risk, whilst the more comprehensive approach may still be better at distinguishing between risk categories among already identified clinical subjects. For use in public health and population triage, it may be desirable to have higher sensitivity, and allow for a higher false positive rate, in order not to miss possible subjects of our cohorts for the planning of preventive efforts.

Both our approach, as well as the approaches of Ahn and Vassileva [34] and Stamates, Linden-Carmichael, Preonas, and Lau-Barraco [11], should be included in future studies of craving and relapse in longitudinal and real-time studies of both non-treatment seeking and clinical populations, as well as in population-level risk triage. The prevailing approach is often to use screening tools such as the Alcohol Use Disorder identification test (AUDIT) or CAGE to identify subjects at risk. Our approach is directed towards potentially identifying at-risk subjects earlier before progressing into actual alcohol dependence, since ADHD symptoms in particular and neurocognitive dysfunction in general have been shown to be precursors of later deterioration with respect to alcohol and substance abuse trajectories (see, e.g., [16])

We did not use binary diagnostic or clinical cut-offs for the predictors within mental health or neurocognitive symptoms but instead analyzed them as continuous variables explaining the probability of the binary CAGE2 score. Albeit not unique to this study, we consider this being a valuable contribution to the existing literature, especially in a Nordic context. Studying populations based on clinical cut-offs or categories limits the results to subjects with high symptom load and might not sufficiently reflect the dimensionality of the phenomena in interest. The presence of executive dysfunction in the current age cohort may contribute to the development of alcohol dependence later in life. It also moderates the expression of internalising symptoms on high-risk alcohol use without necessarily reaching clinical thresholds this early. In addition, such a dimensional approach may help in the conceptualization of the basic aetiology of addiction disorders. We therefore consider not using clinical cut-offs on common psychometric instruments as a strength of our study and possibly a useful contribution to public health efforts in this population.

### Limitations and Further Research

The fact that the study it is cross-sectional obviously prevents us making causal inferences. However, we argue that internalising and externalising symptoms in the form of executive dysfunction, anxiety, and depression are transdiagnostic proxies of other aggravating and protective factors that may influence the odds of being at risk for current or later alcohol dependence. The current data were obtained through a population health survey, which is a strength to this study, though it should be noted that the study suffered from significant attrition, especially of male subjects, which may limit generalizability and create bias, in particular based on sex. We did not include background variables such as socioeconomic status in our study. Although these are available for use in HUNT, we consider them being out of scope for this particular study. Previous findings [16] lend support to this approach, as ADHD remained an important predictor of alcohol use disorder even when controlling for, amongst other factors, socioeconomic status.

Self-reported executive dysfunction was indirectly measured by assessing symptoms typically associated with ADHD. We have not been able to identify studies assessing the correlation between the short form of ASRS and measures of executive function. Studies evaluating this validity question are, however, available for the full, 18-item instrument (see e.g., [35]). Additionally, since the short-form version of ASRS is as good as, if not better than, the full version at measuring the underlying constructs [19], we considered it a prudent approach to use the six-item version as a proxy of executive dysfunction. We used a two-factor solution to the ASRS-6 where factor 2 (hyperactivity/impulsivity) had a low internal consistency, so these results should be interpreted with caution. Future studies should include more elaborate and direct self-reported and performance-based neurocognitive function measures, as they likely fill different, albeit essential, roles in predicting real-life functioning [36]. In addition, common trait measures of impulsivity, such as the BIS-11, might be useful.

Further, it has been reported previously that CAGE may overestimate risk in the current population. Hence, a higher cut-off or an ordinal approach, with CAGE values between 0 and 4, could have been employed. This would, however, have resulted in an even more unbalanced dataset.

Future research should also look further at how neurocognitive functions mediate and moderate the prospective effects of other predictive factors such as anxiety and depression on the probability of high-risk alcohol and substance use. As an example, the age cohort under study here could be followed up in any later waves of the HUNT population studies. Validity will possibly also increase by using continuous biological alcohol markers such as phosphatidylethanol, by applying a prospective design and by including a wider range of neurocognitive measures, including standard neurobehavioral measures of cognitive functioning.

Neurocognitive functions also need to be assessed as state dysfunctions indicated by high-frequency assessments in longitudinal designs through different forms of digital symptom mapping and phenotyping, in line with the research of Stamates et al. [11]. Because within-subject variability in impulsivity indicates higher risk levels than between-subject variability, future multimodal and multilevel model approaches, paired with machine learning techniques, may enable us to analyse when the risk is expressed (state), rather than just who is at risk (trait).

## 5. Conclusions

We demonstrate that symptoms of inattentiveness, hyperactivity/impulsivity and anxiety as measured by self-report scales further our ability to classify individuals with or without increased risk of alcohol dependence in a general population of young adults. We believe that this is important in population-level risk assessment and individual-level risk triage, thus facilitating more person-centered approaches to risk management. This may facilitate early identification and intervention for this age group, before progression into actual alcohol dependence, by looking at the scores and relative importance of these measures. Our study contributes to the existing literature by not using clinical cut-offs for predictor variables and by providing a model based on common and available self-report instruments, particularly in a Nordic context. Nevertheless, generalizations based on sex should be made with caution due to the risk of sex-specific attrition bias. Later studies could use more ecological and intensive longitudinal designs to determine when subjects are at risk and not only who are at risk.

## Figures and Tables

**Figure 1 ijerph-18-11601-f001:**
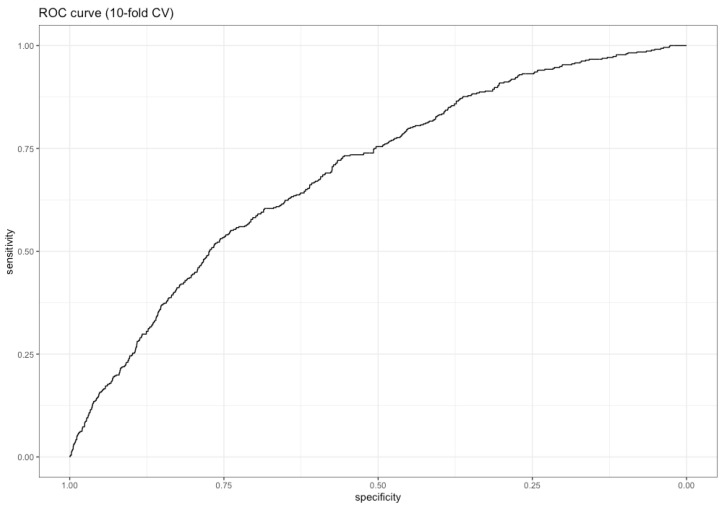
Receiver Operating Characteristics (ROC) curve for the final model, with an AUC of 0.692.

**Table 1 ijerph-18-11601-t001:** Input variables for model selection, *p*-values for two sample *t*-tests for between-sex differences and chi-square for proportions of sex. HADS: Hospital Anxiety and Depression Scale, ASRS: Adult ADHD Self Report Scale—Screener.

	Range	Overall	Male	Female	*p*-Value
Sex		3917 (100%)	1429 (36.5%)	2488 (63.5%)	
Age (years)	19–30	24.9 (3.0)	25.1 (3.0)	24.9 (3.0)	0.04
HADS Anxiety	0–21	5.65 (3.8)	4.89 (3.3)	6.08 (4.0)	7.14 × 10^−23^
HADS Depression	0–21	3.36 (3.1)	3.45 (3.0)	3.31 (3.1)	0.03
ASRS Inattentive symptoms	0–4	1.34 (0.7)	1.33 (0.7)	1.35 (0.8)	0.50
	0–16	5.35 (3.0)			
ASRS Hyperactivity/Impulsivity symptoms	0–4	1.80 (0.9)	1.92 (0.9)	1.74 (0.9)	5 × 10^−9^
	0–8	3.61 (1.8)			

**Table 2 ijerph-18-11601-t002:** Final model with logistic regression output, Odds ratios (OR), and 95% confidence intervals (95% CI) for Odds ratios. HADS: Hospital Anxiety and Depression Scale, ASRS: Adult ADHD Self Report Scale Screener.

	Coefficient	SE	z Value	Pr (>|z|)	OR	95% CI (OR)
(Intercept)	−2.72	0.46	−5.97	2.37 × 10^−9^		
Sex (Male)	1.15	0.11	10.52	7.17 × 10^−26^	3.14	2.54, 3.90
Age	−0.04	0.02	−2.15	0.0315	0.96	0.93, 1.00
HADS Anxiety	0.08	0.02	5.00	5.73 × 10^−7^	1.08	1.05, 1.11
ASRS Inattentive symptoms	0.22	0.07	3.08	2.053 × 10^−3^	1.25	1.08, 1.44
ASRS Hyperactivity/Impulsivity symptoms	0.20	0.06	3.32	9.067 × 10^−4^	1.22	1.09, 1.38

**Table 3 ijerph-18-11601-t003:** Sensitivity and Specificity of the model at three given cut-off thresholds, and the 0.153 cut-off that provided the optimal cut-off (highest sum of sensitivity and specificity).

Cut-off Threshold	Sensitivity	Specificity	Balanced Accuracy
0.05	0.96	0.17	0.57
0.10	0.74	0.53	0.64
0.153	0.55	0.74	0.65

**Table 4 ijerph-18-11601-t004:** Five hypothetical subjects with their predicted probability of having a CAGE score ≥ 2. The optimal cut-off was set at 0.153. The first two subjects approximately achieved this predicted probability.

	27 y.o. Male		28 y.o. Female		30 y.o. Female Low Scores		20 y.o. Male High Scores		20 y.o. Female High Scores	
	Value	Score	Value	Score	Value	Score	Value	Score	Value	Score
Intercept		−2.72		−2.72		−2.72		−2.72		−2.72
Sex (Male)	1	1.15	0	0	0	0.00	1	1.15	0	0.00
Age	27	−1.08	22	−0.88	30	−1.20	20	−0.80	20	−0.80
HADS-A	5	0.4	8	0.64	5	0.40	20	1.60	20	1.60
ASRS IA	1.5	0.33	3	0.66	1	0.22	3	0.66	3	0.66
ASRS H/I	1	0.2	3	0.60	2	0.40	4	0.80	4	0.80
Predicted probability of CAGE ≥ 2		0.152		0.155		0.052		0.666		0.387

## Data Availability

The Trøndelag Health Study (HUNT) has invited persons aged 13–100 years to four surveys between 1984 and 2019. Comprehensive data from more than 140,000 persons having participated at least once and biological material from 78,000 persons are collected. The data are stored in HUNT databank and biological material in HUNT biobank. HUNT Research Centre has permission from the Norwegian Data Inspectorate to store and handle these data. The key identification in the data base is the personal identification number given to all Norwegians at birth or immigration, whilst de-identified data are sent to researchers upon approval of a research protocol by the Regional Ethical Committee and HUNT Research Centre. To protect participants’ privacy, HUNT Research Centre aims to limit storage of data outside HUNT databank and cannot deposit data in open repositories. HUNT databank has precise information on all data exported to different projects and are able to reproduce these on request. There are no restrictions regarding data export given approval of applications to HUNT Research Centre. For more information see: http://www.ntnu.edu/hunt/data (31 January 2020). Data was provided 31 January 2020.

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
