# Peer review of "ADHD and Mental Health Symptoms in the Identification of Young Adults with Increased Risk of Alcohol Dependency in the General Population—The HUNT4 Population Study"

_ijerph, 2021, doi:10.3390/ijerph182111601_

Round 1

Reviewer 1 Report

Authors' aim was to identify factors that could explain the probability of being at risk for alcohol dependence. The article is interesting and well-written, but I have some commnets mainly on the structure of the paper and the methodology of the study. 

First of all, I believe that the introduction is quite long and needs a different structure. Authors focus on the previous literature on executitve functions (and ADHD) and alcohol abuse and one could think  that this would be the aim of the current study as well. However, authors report that their aim is to investigate the factors associated with alcohol dependence in the normal popoulation. So why dedicate so many paragraphs on ADHD and executive function? Maybe they should state in a clearer way that they investigate the alcohol dependency in relation to internalising and externalising symptoms...

Otherwise, if they state that they intend to investigate the factors that in are generally associated with alcohol dependency, then they should include more factors that those included in the current study. Moreover, I would suggest the word "general" instead of "normal" population. Otherwise, please mention what do you mean by normal? 

Authors state that there are meta-analyses that have found that ADHD and impulsivity has been associated with alcohol dependence. If so, what the current study adds to the literature? Authors should say more on the novelty of the current study. 

In methods, authors refer to how they created the variable CAGE2, but I am not sure if this is necessary for readres. Stating that a score more than 2 was used as a cut-off is enough. However, as a reader I would be more interested in knowing which questions are included in CAGE. What do they assess? 

In the results I would expect authors to report more on their main findings (summarized in table 2)-maybe report more on odds ratios for each factor (age, sex, ASRS Inattentive symptoms, etc).

In conclusions I would also expect a more meaningful interpretation of the results and maybe a clinical implication

Line 197: there is a repetition of "3. Results"

Reviewer 2 Report

The article describes using self reported ADHD data and anxiety and the authors determine that they are predictive of alcohol abuse. The article is well written but I have a few comments and suggestions.

1) The cohort used (HUNT4) has substantially more females than males, but it was not given consideration when other generalizations were made, i.e. "Female subjects had higher average anxiety (p>0.001) and depression scores (p>0.05). Whereas male subjects had higher average Hyperactivity/Impulsivity scores (p>0.001)". This could be due to the bias of the cohort unintentionally excluding males that fit the criteria that's result in such sex differences, e.g. male subjects with higher anxiety tend not to want to participate and sign up for the study, i.e. sampling bias. As such, any generalization between male and females should be interpreted with caution. Also, the the p-values here should be <, not >.

2) I find that the CAGE score >=2 to be somewhat arbitrarily chosen. It is also not clear to what extent the answers to the 4 questions are correlated. Perhaps the authors can try the model using other criteria, e.g. score >=1 or score >=3 or test the different questions independently and see to what extent the conclusions they reach remains the same, better or worse?

3) In Table 2, the Pr(>|z|) is given as a general <0.01 or <0.001. I was wondering why don't give the exact values instead? It can be misleading as ASRS Inattentive symptoms ASRS and Hyperactivity/Impulsivity symptoms seem to have similar results yet one is <0.01 and the other is <0.001, which suggest a 10-fold difference. Also, I'm not sure what's the point of showing the intercept here? Seems redundant.

4) I was wondering what's the statistic for testing depression? The authors clearly have the data for it, but did not show the result. Even if it were not significant, I think it would be great to show the result of the test.

5) I wasn't quite sure how the cut-off threshold was calculated until I saw table 4 where it is the Predicted probability of Cage ≥ 2. This suggest it is not calculated at all but they used their model to get a statistic for every sample in the cohort and base on that threshold, determine the probability, i.e. how many had Cage >=2. Perhaps, the authors should elaborate more about this in the methods section.

Reviewer 3 Report

I can appreciate the value of having longitudinal data on a representative sample of the 'normal' population. However this research methodology faces major challenges in terms of sample attrition, available measures and low frequency occurrence of particular conditions relevant to public health.  To my mind, this paper fails to meet these challenges.

Given the main focus of the study, it would be preferable for the authors to investigate the psychometric properties of the ASRS-6 scale with the sample used in their study.  Hopefully this would replicate the data reported for  the scale's previous use in terms of the two factor structure but their analysis  should  add some important additional information, notably the percentage of variance accounted for by the two factors. The Cronbach alphas suggest a weak coherence in the hyperactivity/impulsivity factor and again, seeing the factor structure for these two items could be illuminating.   It might be preferable to use  these items separately in the subsequent analyses assuming that collinerarity is not an issue.

For the three scales used in the study it would be helpful if the authors could report the test-retest reliabilities As reported previously, the authors do not report the intervals between the data gathering for the HUNT study but is there not an opportunity to report the correlations on the three chosen measures from HUNT3 and HUNT4?  Furthermore is there not the possibility of replicating their analyses with other waves of the longitudinal study?  Such analyses would serve to demonstrate the robustness of the relationships they identify using the HUNT4 and extend them by noting any changes on the measures across the two waves.

Although the authors drew on a longitudinal study in one Norwegian region, the final sample used in these analyses was less than 3,917 ( due to missing data).  The authors fail to report the final numbers used in the analyses but in any case, this is less than a 25% sample and may  contain additional biases from the population characteristics.  Indeed, the  authors, note that attrition in HUNT4 was significant by gender and possibly on other indicators such as socio-economic status, educational level and rurality. The authors should clarify and discussion further within the limitations.

Linked to this for an international readership, it would be helpful to know about the availability and use of alcohol in Norwegian society and if there is comparative usage data from other countries. The authors should acknowledge that their findings at best are applicable to Norway and need to replicated in other transnational studies.

The authors do not report the amount of variance accounted for by the CAGE2 regression model. I also wonder why other social predictors (see above) were not included in this analysis.

The ROC statistics are poor and do not meet the sensitivity and specificity indicators usually expected for screening tools. Moreover the proposal to use a score of 1.53 is illogical given the nature of the scoring on this indicator.  Rather the authors need to report these ROC statistics for the cut-off of 2 which is the basis for their regression analysis.  In summary, these analyses suggests that CAGE2 is not fit for purpose and other screening tools need to be developed.

The authors need to provide greater justification for the data presented in Table 3. I fail to see its value.

The summary of the findings in the discussion are overstated given the small odds ratio reported with confidence intervals close to 1.00.  Indeed I fail to see how these findings could be confidently translated into useful tools in public health and primary care.

As noted above, the study has many limitations and no amount of statistical analysis can overcome poor quality data on which they are based and possibly an inadequate sample.  No doubt the study proved a valuable learning experience for the doctoral student which is a worthy outcome.  Equally informative would a critique of the limitations of 'big data' studies to explore complex relationships which occur with low frequency in populations and their failure to include relevant and robust measures.

Round 2

Reviewer 2 Report

The authors performed necessary changes and edits. I have a couple of minor points/suggestions,

The authors decided to put the p-values directly but the Sex variable in Table 1 and 2 had the p-value listed as p<2 x 10^-16 instead of the actual p-value. This is probably due to R showing this in their summary. The authors can extract the actual p-value by accessing the variable directly, i.e. variable_name$p.value

As for the dropping of the "HADS Depression" variable from your model because of AIC, I understand the process, but it does question why was HADS Depression dropped. The authors had 6 variables in Table 1, but only 5 variables in Table 2. Only HADS Depression was dropped. The question is whether it was dropped because it is not associated with ADHD or that it is associated but it does not add any predictive power to the model and thus was dropped. My comment was mainly to figure this out.

Nonetheless, I thank the authors for the insightful work and look forward to it being published.

Thanks!

Author Response

Dear reviewer 2, thank you again for your feedback. Here are the responses to your suggestions.  

  • The authors decided to put the p-values directly but the Sex variable in Table 1 and 2 had the p-value listed as p<2 x 10^-16 instead of the actual p-value. This is probably due to R showing this in their summary. The authors can extract the actual p-value by accessing the variable directly, i.e. variable_name$p.value

Response:

Indeed that is the reason, and exact p-values have now also been added to tables 1 and 2 as well as in line 201.                        

  • As for the dropping of the "HADS Depression" variable from your model because of AIC, I understand the process, but it does question why was HADS Depression dropped. The authors had 6 variables in Table 1, but only 5 variables in Table 2. Only HADS Depression was dropped. The question is whether it was dropped because it is not associated with ADHD or that it is associated but it does not add any predictive power to the model and thus was dropped. My comment was mainly to figure this out.

Response:

We think that you are referring to CAGE2 as the response variable. CAGE2 was weakly correlated to HADS Depression (0.102, df=3775, p= 8.55 x 10-11). The model selection was performed by choosing the model with the lowest value of the AIC among all possible additive models. The ‘HADS Depression’ was then not inclueded in the model. The AIC makes a tradeoff between model complexity and model fit. We have added a sentence about  this matter in lines 219-221 in the revised manuscript.

Reviewer 3 Report

Your response to my comments underline the reservations I have about your study.  I appreciate that you are unable to respond to some of the important issues I raise but there are ones which you seem reluctant to acknowledge such as the poor psychometric properties of the key indicator ASRS-6.   As the limitations to your study are many - not least sample attrition - I cannot see any  value in publishing it in the Journal as it will be of little benefit to other researchers and clinicans. 

Author Response

Dear reviewer 3, thank you again for your work on your manuscript, that we believe has improved it. We are sorry that you still do not see any value of publishing it. In the previous resubmission of the manuscript, we have tried to address some of the value and contribution we believe it adds. All the while acknowledging your concerns with respect to attrition, generalisability and psychometric properties of the ASRS-6 instrument. Without trying to dodge any of your concerns.